# Music Therapy Assessment for Older Adults: Descriptive Mixed-Methods Study

**DOI:** 10.3390/bs14050354

**Published:** 2024-04-23

**Authors:** Amy Clements-Cortés

**Affiliations:** Faculty of Music, University of Toronto, St. George Campus, 80 Queens Park, Toronto, ON M5S 2C5, Canada; a.clements.cortes@utoronto.ca

**Keywords:** music therapy, assessment, older adults, survey, domains of health

## Abstract

Background: The purpose of this inquiry was to test the new ‘Music Therapy Assessment for Older Adults’ (MTAOA) tool in Canada and the United States, and to establish its content and predictive utility. Methods: A pilot study using an explanatory descriptive methods design was chosen; *n* = 18 music therapists completed an online survey about their experiences in administering the assessment and 50% (*n* = 9) were invited for a follow-up interview. Results: The results indicated that the MTAOA was a beneficial assessment tool that contained relevant domains (89%) to develop a music therapy treatment plan; 89% of music therapists also noted they would continue to use and recommend the MTAOA. The data produced beneficial information that were used to revise the assessment form to ensure inclusive language and reduce any potential inherent or unconscious biases. Conclusions: Future research is needed to assess the utility of the revised MTAOA in other global regions where music therapists work with older adults.

## 1. Introduction

Music therapy is defined as: “…the professional use of music and its elements as an intervention in medical, educational, and everyday environments with individuals, groups, families, or communities who seek to optimize their quality of life and improve their physical, social, communicative, emotional, intellectual, and spiritual health and wellbeing” [1]. The use of music-based interventions (MBIs) that include music therapy are increasing and demonstrate benefits for health and wellness including the treatment of neurological disorders [2]. The National Institute of Health recently published a music-based interventions toolkit in order to advance the development of studies assessing the impact of MBIs for persons with dementia [2], drawing attention to the important implications for the public to understand and advance music therapy and MBIs in work with older adults. One way to advance the application of music therapy interventions is to further develop appropriate assessment and evaluation tools. 

Music therapy assessment has received increased attention over the past few years with the publication of several books as well as new assessment tools for various populations and/or clinical needs [3,4,5,6]. Further, an International Music Therapy Assessment Consortium https://www.musictherapy.aau.dk/imtac (accessed on 10 October 2023) between Aalborg University, Chroma, Molloy College, National Hospital of the Faroe Islands, Temple University, University of Applied Sciences of Wurzburg, University of Jyväskylä, and University of the Pacific [7] was formed to advance music therapy assessment and to serve as a repository for music therapy students, clinicians, and researchers to locate assessment tools for use in their practice.

There are still relatively few standardized assessment tools within the field of music therapy and those developed are often directed towards a specific diagnosis or population [8]. The American Music Therapy Association’s (AMTA) Standards of Clinical Practice [9] considers assessment a fundamental component in providing music therapy treatment [6,10]. 

Music therapy assessment can be defined as a structured process of 

(1)preparation,(2)data gathering,(3)analysis, interpretation, and conclusions about the assessed information,(4)documentation and communication of musical and non-musical data about the music therapy process in order to provide information to make decisions, raise hypotheses, get to know clients better, and achieve a better understanding of the music therapy process ([4], p. 15).

Music therapists work with persons across the lifespan and have the unique opportunity to contribute to care plans and collaborate with other disciplines. Effective assessment tools afford a holistic view of the individual, create an effective treatment plan, and share important information and observations with other clinicians (where applicable) to foster increased understanding of the scope of music therapy praxis. With the heightened number of older adults in society [11], there is likely to be an increase in music therapy provision to this demographic, and it is timely to look at developing music therapy assessment tools accordingly. At present, there are limited music therapy tools designed to consider the varying needs of older adults; in particular, there are a lack of assessment tools that are culturally relevant and free from assumptions and biases of abilities. Many music therapy assessment forms currently used in practice are generic tools used for individuals across the lifespan that may have been developed from other health care professionals’ tools [3]. While the same domains may be assessed, an assessment tool that includes items specifically tailored to the needs of older adults is warranted.

## 2. Literature Review

Effective assessment tools should include non-musical (psychological, cognitive, physiological, and communication) and musical (musical skills, responses, and preferences) domains, and reflect current evidence-based practice [8]. Additionally, the reason for referral, the client’s current condition, and an intervention plan directed towards desired outcomes are necessary parts of a robust assessment tool [12]. According to Wigram et al. [13], music therapy theory has been developed largely through empirical practice, resulting in less emphasis being put on developing standardized assessment protocol. Due to the lack of research and the creation of standardized assessment tools in music therapy, many music therapists have become accustomed to developing their own assessment tools [14], or adapting standardized assessments from other related disciplines, often compromising the validity and reliability of the tool [12].

Comprehensive geriatric assessment (CGA) involves assessment of cognitive, affective, environmental, spiritual, financial, and social influences on a person’s health [15]. Major components assessed in the CGA are functional capacity, fall risk, cognition, mood, polypharmacy, social support, financial concerns, goals of care, and advance care preferences [16]. While some of these domains are beyond the scope of practice for a music therapist to assess, they do inform the importance of looking at and assessing older adults in a holistic manner. Wister [11] notes that important domains to assess in older adults include physical, psychological, spiritual, social, and functional.

Standardized and clinician-developed music therapy assessments have been created for older adults with varying needs, including: general assessments for older adults [17,18,19], dementia (Music in Dementia Assessment Scale: MiDAS) [20,21], Huntington’s disease (Music Therapy Assessment Tool for Advanced Huntington’s disease) [10], disorders of consciousness (MATADOC) [22,23,24], and acquired brain injury [25,26]. Three examples are described below. 

The Magnet Assessment (MAGNET) is a tool that does not require specialized training and was created for assessing older adults in long-term care settings. The assessment was developed to include the domains that are federally mandated in the United States as part of assessment, such as cognition, communication, mobility, etc. Interviews, observation, and music engagement are part of the assessment to evaluate client functioning and determine the therapeutic process [17]. 

The Music-Based Assessment for Cognitive Function for Adults with Acquired Brain Injury (ECMUS) was developed by Pfeiffer et al. [25,26]. It is intended to assess cognitive function, has four subscales—attention, memory, executive functions, and mood—and, at present, is available in the Spanish language. The assessment involves singing, improvising, identifying, and tracking sounds. It comes with a user manual and the required visual and audio files [27]. It is administered over 1–2 sessions that are approximately 50 min in duration. The purpose is to serve as a baseline for treatment [27]. 

The Music Therapy Assessment Tool for Awareness in Disorders of Consciousness (MATADOC) is available in the English and Spanish languages [22,24,28]. It is used with individuals across the lifespan. Looking at three subscales, two scores are produced to determine the level of consciousness and musical responsiveness, as well as the clinical goals and interventions. The MATADOC assessment is done over four individual music therapy sessions over an 8–10 day period, and various musical tasks are completed, observed, assessed, and scored [29]. This tool requires advanced training to administer and evidence has been published regarding its validity and reliability [30]. 

However, music therapy research lacks assessment tools for older adults who may present with a variety of health issues as part of the life course. A thorough assessment sets the baseline and the start of the treatment process and deserves increased attention. The absence of a standardized and externally validated music therapy assessment is a limitation within the field [31]. According to Jacobsen [32], this is a methodological issue that can be resolved through conducting further pilot studies and clinical trials. 

## 3. Purpose of the Study

The purpose of this study was to create and pilot a comprehensive domain-based music therapy assessment tool that can be used with adults aged 60 years and above. The information gathered from the results was used to design the revised Music Therapy Assessment for Older Adults (MTAOA). The revised MTAOA can now be taken to other experts working in gerontology to gain further insights into the final domains, scales, and descriptors. The MTAOA could then be piloted to a larger sample of music therapists in a future study. There is general agreement within the research literature that pilot studies are a useful method for trialing a study design and for testing the suitability of the research instrument [33]. The aim was to establish content and predictive utility of the Music Therapy Assessment for Older Adults (MTAOA) for older adults and its applicability in assessing a variety of issues that might impact older adults (See Appendix A).

## 4. Materials and Methods

The pilot study utilized an explanatory descriptive [34] mixed-methods [35] design [36] where 

participants completed an online survey to evaluate the MTAOA, and50% of participants were invited to participate in a follow-up interview based on purposive sampling. Interview questions were designed after analyzing the survey data to understand survey responses in a more fulsome manner. Descriptive survey research is aligned with assessing the experiences of music therapists [34]. The mixed-methods approach was chosen to gather initial feedback on the questionnaire-based survey and followed up with an interview taking a guided interview approach [37] where the interviewer could probe and/or seek clarification on responses [38].

### 4.1. Research Questions

What are the essential components/domains of a standardized music therapy assessment tool for older adults? 

What are the most effective word descriptors to include in the various domains?

What is the viability and applicability of the MTAOA assessment tool with older adults in Canada and the United States?

### 4.2. Assessment Tool

The MTAOA piloted in this study (See Appendix A) was developed by the Principal Investigator (PI), and two former students of the PI both working with older adults in private practice and in long-term care. Further, music therapists working with older adults (*n* = 23) were asked to review drafts of the MTAOA and provide informal feedback with respect to the layout, domains, descriptors, and viability. Older adults were also consulted on the domains and descriptions as well as the applicability of the tool overall. The theoretical framework that underpins the assessment is the view of aging as a social process [11]. 

“Aging, as a social process, involves multi-level and complex interactions between individuals and various social structures and systems; within changing social, economic, political, policy, and physical environments; and across diverse cultural contexts, all of which vary…across one’s life course”.([11], p. ix)

The assessment form was also designed after reviewing other published music therapy assessment tools [17,19,20,21,25,39,40,41,42,43] and former assessment tools utilized by the MTAOA creators in their work settings, and from the PI’s experience working with older adults for 25 years. This review involved assessing the domains, items, and scales that others have included alongside terminology and descriptors. Alongside this review, a literature review on geriatric assessment was conducted to further inform the areas to be assessed on the MTAOA. 

The MTAOA gathers background and demographic information; conducts assessment in cognitive, communication, psychosocial, physical, and musical domains; and uses a dropdown menu. Some areas may be beyond the scope for a music therapist to assess, such as aphasia, but this information may be available to the therapist from the chart or client/consumer/service user and/or substitute decision makers and can therefore be included and reported on the MTAOA. There is an area for a narrative summary and comments on each of the domains assessed. Initial goal areas are presented in a drop-down menu and there is a space to note other goals. 

The form is intended to be used as an initial assessment and planning treatment and could be used in individual or group settings. It includes the use of interviewing and observing, and involves various music therapy interventions such as singing and playing instruments. A reading card is needed for one item on the assessment. There is no set time that the assessment requires, and the assessment can range from 30–60 min depending on the energy, attention, and cognitive levels of the older adult as well as the environment where the assessment is conducted. 

### 4.3. Recruitment

A total of 8–20 music therapist participants were proposed to participate in this study. Participants were recruited via purposeful sampling as well as through e-mail invitations. E-mail invitations were sent to potential participants via the Canadian Association for Music Therapists as well as through direct e-mail invitations from the PI and Research Assistant (RA) and postings on social media groups seeking participants who resided in Canada or the United States. Recruitment spanned from November 2019 to April 2022. As a result of the COVID-19 pandemic and restrictions placed on research, the study was on hold for a brief period in 2020 as per Research Ethics Board requirements. Only two participants were recruited in 2020, primarily due to many music therapists indicating they desired to participate but their caseload had changed because of the COVID-19 pandemic restrictions. Potential participants contacted the RA, who provided them with the study consent form. Upon signing the consent form, the MTAOA was sent to the participants with instructions. The RA was also responsible for sending out the survey link to participants when they were done using the assessment form. 

### 4.4. Participants

Participants included credentialed music therapists (MTA) working in private practice (whose practice involved at least 50% of their caseload focused on work with older adults) in Canada, and Music Therapists Board Certified (MT-BC) working in the United States with older adults who had either an undergraduate or graduate degree in music therapy. Participants had to have worked with older adults for over three years and provided both individual and group music therapy programming. All participants provided contractual music therapy services to older adults in long-term care, retirement, hospice, community, and private homes and worked with older adults who came from a variety of racial and cultural backgrounds. A total of *n* = 23 participants were recruited, 2 withdrew and 3 did not respond to the follow up e-mail requests to complete the survey. A total of *n* = 18 participants completed the survey, and 9 interviews were conducted. The participants ranged in age from 23 to 57 (1 male, 2 non-specified and 15 female) and their work experience with older adults ranged from 2–26 years with an average of 9.9 years. The racial and cultural background of the participants was not collected. The 2 participants who withdrew expressed they were not able to use the MTAOA after consenting to participate in the study given the client/consumer/service user currently in their private practice. Only 1 participant knew the RA in this study. Out of the *n* = 18 participants who completed the study, 10 knew the PI in some capacity, i.e., as a colleague, former student or from meeting them at a professional association conference. These participants the PI had previous relationships with were interviewed by the RA and all correspondence with them went through the RA. This was intentional so participants would feel they could be truthful in sharing their experiences and perceptions and to reduce any potential bias. 

### 4.5. Procedure

Participants were asked to use the MTAOA assessment form created for the study to conduct individual or group music therapy assessments with new older adult client/consumer/service users in their private practice with a variety of health diagnoses and issues at the beginning of the therapeutic process. They were asked to conduct a minimum of five assessments with no maximum number of assessments required. Upon completion of this part of the study, participants contacted the RA to indicate they were ready to complete the survey on the utility of the MTAOA (See Appendix B). Participants did receive periodic e-mail correspondence from the PI and/or the RA to check in on their progress. When all the participants had completed the survey and data were analyzed, 10 participants were randomly invited to participate in a follow-up interview and 9 out of 10 completed the interview. It became challenging for the final participant to complete the interview given their schedule and that of the PI interviews took place over the Zoom platform and lasted approximately 30–45 min. Interviews were recorded for transcription and verified for consent with participants. (Please see Appendix A for the interview questions).

Clients/consumers/service users were asked for informed consent to participate in music therapy (standard practice) and this included an explanation that an assessment would be conducted at the beginning of the therapeutic process. Clients/consumers/service users were not considered participants in the study as their data were not being analyzed. 

### 4.6. Data Analysis and Trustworthiness

Data were collated from the survey and responses tallied. A description of the results was produced. The results of the survey informed the questions that comprised the interview questions. A thematic analysis [44] of the interview data was conducted to organize primary codes and themes [45]. Braun and Clarke’s [44] six-phase process and guidelines for thematic analysis (familiarizing with the data, generating initial codes, searching for themes, reviewing themes, defining and naming themes, and producing the report) was selected integrating Joffe and Yardley’s [45] coding guidelines. Member (participant) checking and peer debriefing were completed to ensure the interview responses reflected the participants’ expressions [46]. Triangulation was provided by collecting both survey and interview data, and constructs were allowed to emerge from the analysis [47].

The method chosen for this study was appropriate to the purpose, thus adhering to criteria for trustworthiness in research as outlined by Lincoln and Guba [47]. The researcher approached the data analysis with a desire to learn of the experience of the therapists and their assessed benefits, challenges, and changes to the assessment form. This is also why the researcher chose to include music therapists with varying levels of work experience with older adults to ensure generational and other perceptions were represented as well as to ensure transferability of the utility of the MTAOA in various settings where participants worked.

### 4.7. Ethical Considerations

Research Ethics Board approval was obtained from University of Toronto. Participants were assigned a study number to ensure confidentiality and the list was randomized. The consent form indicated the participants could withdraw from the study at any point. The list of participant names, interview audio recordings, and transcriptions were discarded after participant verification. Surveys were discarded upon publication of the research study. All electronic files were kept on the PI’s password protected laptop as well as the University secured OneDrive.

## 5. Results

### 5.1. Survey

#### Requested Additions and Changes

There were several suggestions for additions and changes to the MTAOA. With respect to background information, the following changes were proposed to be added: Gender identity 5% (*n* = 1), pharmaceutical information 5% (*n* = 1), prior music therapy involvement (*n* = 1), preferred pronouns 5% (*n* = 1), a place to note if consent was provided by facility 5% (*n* = 1), and date of birth 5% (*n* = 1). A terminology replacement was raised from using “ethnic background” to cultural information 5% (*n* = 1). The only item that was suggested to be removed was education 17% (*n* = 3).

### 5.2. Domains

Suggestions in the cognitive area included adding a place for standardized test scores such as the Montreal Cognitive Assessment [48], more specific instruction on following directions, i.e., 1-step or 2-step 11% (*n* = 2), a place to list the amount of time attention was sustained 5% (*n* = 1) and a standardized card with the sentence to be read by the client/consumer/service user during the assessment to ensure the same font size was used and the text was clearly written and legible 5% (*n* = 1). Three participants suggested changing terminology surrounding the “ability to” language in the assessment overall, proposing to reword or remove those words and rephrase the statements. For example, “ability to choose” could be reworded as client/consumer/service user chose between two items, or client/consumer/service user did not choose between two items.

In the communication domain, additions of intelligibility of speech 17% (*n* = 3), stuttering 5% (*n* = 1), hearing ability 5% (*n* = 1), and use of sign language and other communication systems or devices 17% (*n* = 3) were suggested. Additions to the psychosocial domain were to include understanding of emotions 5% (*n* = 1), coping skills 5% (*n* = 1), and 5% strengths (*n* = 1). Changes suggested included removing subjective words such as pleasant and engaged and replacing these with notes about history of aggression, anxiety, etc. 5% (*n* = 1). Two participants suggested removing “inappropriate emotional expression” as this could be captured in the other area of behavior. Under affective responses, 5% (*n* = 1) suggested adding a FACES scale to assess these responses and no request to remove items was shared.

In the physical area, participants desired more specific information on gross and fine motor abilities including ability to cross the midline of the body and ability to transfer. Further, the use of wheelchair and/or bed bound were suggested to be added. Musical domain requests included adding the ability to sing accurate lyrics 5% (*n* = 1) and ability to improvise 5% (*n* = 1).

All participants found the summary area on the assessment form important and helpful. With respect to the goals listed as examples, participants 94% (*n* = 17) felt this was an appropriate list and that they could add their own goals as needed. One participant asked to change the terminology regarding “dysfunctional behaviors” as well as “spiritual support”.

### 5.3. Content Validity and Predictive Utility

The questions related to content validity and productive utility asked participants to rank their scores from 0–5 (0 = Completely does not describe and 5 = Describes completely). Percentages reported here are for combining rankings of 4 and 5. Overall, 89% of participants reported the assessment form contained all the relevant domains to establish a music therapy treatment plan and the information presented on the form was relevant to music therapy practice. Further 89% (*n* = 16) noted they would both use and recommend this assessment form, and 89% (*n* = 16) also felt it was suited for persons with dementia and acquired brain injury as well as with older adults overall. Eighty-two percent noted the assessment would help them determine if a person was suitable for music therapy, and 78% felt the form collected sufficient background information. Eighty-four percent felt the assessment tool was easy to administer, and 89% (*n* = 16) noted the information was clear. All participants agreed that the recommended services section should remain.

### 5.4. Interview

All participants who engaged in the interview (*n* = 9) used the MTAOA with older adults as well as older adults having specific issues including dementia 100% (*n* = 9), acquired brain injury 56% (*n* = 5), post-stroke 33% (*n* = 3), aphasia 33% (*n* = 3), depression 78% (*n* = 7), and Parkinson’s 22% (*n* = 2), among other issues, as well as with some other demographics including children. In addition, 100% of participants (*n* = 9) stated this assessment was appropriate for older adults at large and those with various health issues; while 56% of participants (*n* = 5) felt the MTAOA was also useful for others including adults with mental health needs, children, and adolescents but with some of the categories altered to be more specific to different ages groups or issues. On average, participants used the MTAOA with 8 new clients in their practice before taking the research study survey, and all participants stated they were continuing to use this tool alongside other assessments in their practice.

Participants used the assessment form over one to three sessions to establish goals and objectives and establish a treatment plan depending on the client and the location of the assessment. Assessments were conducted in the client/consumer/service user’s homes, private practice locations, private retirement homes, and long-term resident rooms, as well as via telehealth.

The format of the MTAOA was desirable by all 9 participants with respect to having checkboxes as well as spaces for description and elaboration. There were no specific parts of the assessment that were identified as more important or needed then others, as all participants discussed the importance of gathering as much information as possible to form an accurate assessment of the client/consumer/service user and their needs and desires out of the therapeutic process.

Participants were divided among whether standardized assessments used in healthcare (i.e., MOCA [48], Trail making test [49] should be as part of music therapy practice), and this seemed to be impacted to some degree by the music therapy approaches that informed the participants’ practice. Participants were invited to discuss terminology and language choices used on the assessment form and suggest potential changes. Discussion and responses aligned with survey responses in terms of making the language more inclusive and reducing bias, with less focus on items that might not be readily observable or were highly subjective such as the term ability. When asked how participants refer to the persons they serve, there was again division on what is the best terminology. All participants felt the use of client/consumer/service user was respectful and had no further suggestions on how to further empower those with whom we work.

## 6. Discussion

The MTAOA includes the essential domains of assessment (psychological, cognitive, physiological, communication, music, reason for referral) as noted earlier by Douglas [8,12] to form the baseline for the treatment process and space to record initial goals. While previous assessment tools for older adults are available, they may require training, such as the ECMUS [25,26], and take a long time to administer, such as the MAGNET [17]. Music therapists often have limited time for assessment in their work with older adults and need access to an assessment tool that can be accurately completed in a concise time that gives them a holistic picture and baseline from which to move forward. The MTAOA was received as such a tool by participants in this study.

In ensuring we are practicing from an anti-oppressive framework and with cultural humility and competence, it is essential we look at the language and terminology we are utilizing in assessment. In the past few years there has been increased discussion surrounding the language used in therapeutic contexts, which includes terminology and language on music therapy assessments to ensure biased or highly subjective writing is limited or removed. For example, Webb and Swamy state “...conscious and unconscious bias, as well as invisible barriers, have prevented music therapy scholarship and publication from being as inclusive and anti-oppressive as it should be” ([50], p. 100). One example is with respect to how we refer to the persons we serve. The term client is frequently used alongside patient when referring to persons we work with in music therapy. There is no consensus in the literature on the correct or preferred term to use, and guidelines from the American Psychological Association impart: “It is understood that psychologists will respect individual and/or cultural preferences expressed by recipients of psychological services and their families when choosing language to describe those individuals, families, or populations” ([51], p. 2). In the revised version of the MTAOA, the term client was changed to Client/Consumer/Service User to be more inclusive. This is one small step to help dimmish power differences and opt for the use of client/consumer/service user preferred terminology.

While the use of standardized assessment tools has advantages in terms of ensuring there is consistency in what is assessed and the goals that might be implicated, they have the potential to place individuals into broad overarching groups and pathologize their care in place of offering individualized assessments to support care plans. Khan [52] shared that healthcare professionals operating with cultural humility “…must listen with interest and curiosity, have an awareness of their own possible biases and attempt a non-judgmental stance about what they hear, and recognize their inherent status of privilege as a provider and be willing to be taught by their patients”. A “culturally competent” provider needs to have knowledge and awareness of:health-related beliefs, practices, and cultural values of diverse populations; illness and diagnostic incidence and prevalence among culturally and ethnically diverse populationstreatment efficacy data (if any) of culturally and ethnically diverse populations”

Although the revised MTAOA will include a space to list standardized tests when relevant or applicable, the use of the narrative description categories from the original MTAOA, in addition to the checklist-style form, are offered to ensure that the assessment does not ask music therapists to “box” their client/consumer/service user into one predetermined criteria and rather allow for free writing to ensure assessment is personalized. By not using a more formalized protocol or lists of tasks, this also allows the music therapist the ability to include client-preferred, culturally relevant tasks, instruments and music in assessing the older adult.

While three participants did state they thought education could be removed from the background information, the decision to keep this in the revised version of the MTAOA was based on discussion in the interviews where participants were asked about whether this information was helpful. While, at times, it might not be relevant, education could also provide information that would benefit the interpretation of standardized tests and serve as information to foster reminiscence.

As consent is an important part of the therapeutic process, it was important to add that consent is sometimes not provided by an individual in written form, but rather by a facility. It is, however, essential that music therapists do follow up and seek verbal consent from their client/consumer/service user or assent if consent is not possible.

While not part of this study, the PI has given the MTAOA to students and supervisees to use in their clinical placements over the past three years, and 9/10 students used it in their placements with older adults, so approximately 55/60 students. It would be interesting to do a follow-up study where two music therapists utilized the MTAOA to assess a client and the results were compared to confirm inter-rater reliability of the tool.

## 7. Implications, Limitations, and Future Research

As a result of the findings from this investigation, the MTAOA was revised to include items suggested by participants. (Please see Appendix A for Revised MTAOA-R).

This study is limited by the relatively small number of participants. In hindsight, collecting additional demographic information would have been useful to understand the cultural backgrounds of the participants as well as those of their client/consumer/service user and, therefore, it is difficult to assess the diversity of those included. Further, the tool was only used by music therapists living and practicing in Canada and the United States whose training is likely informed by a Western perspective, thus limiting the external validity of the results being translatable in different contexts, which is an issue with assessment noted in the field [31]. In future research, it would be helpful to assess the utility of the MTAOA with music therapists directly employed in long-term care and retirement homes as well as in community centers and adult day programs. Given the different contexts where assessment can take place and the amount of time allotted for assessment (sometimes determined by a facility), there may be further changes for adaptations that would increase the usefulness and efficiency in practice. Given students also identified informally that the MTAOA was useful, a study involving music therapy interns’ utilization of the form is also warranted.

## 8. Conclusions

Given the rise in the number of older adults in society, music therapists are likely to increase their work with this demographic. This descriptive mixed-methods investigation tested the utility of the Music Therapy Assessment for Older Adults (MTAOA). Eighteen credentialed music therapy participants completed an online survey and 9 follow up interviews were conducted to learn about the participants’ experiences in administering the MTAOA in their private practices in Canada and the United States. As a result of feedback from study participants, the MTAOA has been revised and offers a comprehensive tool to assess older adults and develop an initial treatment plan. Further investigation is needed on the utility of the MTAOA by music therapists employed by long-term care homes and other community settings.

## Data Availability

The raw data supporting the conclusions of this article will be made available by the authors on request.

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
