# Peer review of "Music Therapy Assessment for Older Adults: Descriptive Mixed-Methods Study"

_behavsci, 2024, doi:10.3390/bs14050354_

Round 1

Reviewer 1 Report

Comments and Suggestions for Authors

The contribution is highly relevant. The mixed-method research approach should be emphasised in particular.

line 141 ff Materials & Method 

Please critically review the term descriptive mixed-method design. The term explanatory mixed-method according to Creswell & Plano (already in the literature list) is, as far as I know, more common.

l. 153 Research Questions

The research questions are formulated in very general terms, yet the analyses are focused on MAOA. Please reconsider.

l. 413+418

formal issues (6)

Discussion

It would be interesting to check whether two raters arrive at the same assessment. The inter-rater reliability test should at least be addressed in the discussion.

At the beginning of the article, reference is made to the importance of checking external validity. This should at least be mentioned in the limitations and possible assessment instruments for checking external validity should be mentioned in the discussion.

Author Response

The contribution is highly relevant. The mixed-method research approach should be emphasised in particular.

line 141 ff Materials & Method

Please critically review the term descriptive mixed-method design. The term explanatory mixed-method according to Creswell & Plano (already in the literature list) is, as far as I know, more common.

I added in Plano-Clark, Huddleston-Casas, Churchill & Garrett. (2008). Mixed Methods Approaches in Family Science Research. Journal of Family Issues - J FAM ISS. 29. 10.1177/0192513X08318251 which discuss the explanatory design. I also felt it important to keep in the other reference. Thank you for this suggestion.

  1. 153 Research Questions

The research questions are formulated in very general terms, yet the analyses are focused on MAOA. Please reconsider.

Question 3 was revised to reflect this.

  1. 413+418

formal issues (6)

I am sure what you are asking but I think it has to do with the odd formatting. I have added back the bullets that were in my submission to make this easier to read.

Discussion

It would be interesting to check whether two raters arrive at the same assessment. The inter-rater reliability test should at least be addressed in the discussion.

Thank you, this has been added to lines 443-445.

 “It would be interesting to do a follow up study where two music therapists utilized the MAOA to assess a client and the results were compared to confirm inter-rater reliability.”

At the beginning of the article, reference is made to the importance of checking external validity. This should at least be mentioned in the limitations and possible assessment instruments for checking external validity should be mentioned in the discussion.

This has been added to lines 452-454 noting the external validity of this tool to be able to generalize to other contexts.

Reviewer 2 Report

Comments and Suggestions for Authors

This is a very important topic and the effort to enhance music therapy diagnosis and to make it more inclusive and broader is always a good direction

the literature is good

i am concerned that the study included so few people, ( only 8 as far as i understand?) this brings the methods down) i wonder if you could elaborate on that- this is more suited to interview type study

at the same time, the findings make sense, and sound like good inclusions into the assessment package maybe a more tentative language will keep it as a preliminary research with some suggestions? rather than an overarching assessment/ 

maybe you could add some words about that in the methods?    

Author Response

This is a very important topic and the effort to enhance music therapy diagnosis and to make it more inclusive and broader is always a good direction

the literature is good

i am concerned that the study included so few people, ( only 8 as far as i understand?) this brings the methods down) i wonder if you could elaborate on that- this is more suited to interview type study

There were 18 participants in the study and given it is mixed methods we did interview 9 or 50% of them. I am not sure what to elaborate upon. The design is noted as mixed methods and the interviews were to learn more about the survey results.

at the same time, the findings make sense, and sound like good inclusions into the assessment package maybe a more tentative language will keep it as a preliminary research with some suggestions? rather than an overarching assessment/ maybe you could add some words about that in the methods?    

I am also not sure what to note in the paper. The study is a pilot and this is noted with references to support the design. I am happy to make more changes but am unsure what is being asked.